# Relational models theory: Validation and replication for four fundamental relationships

**Michael Zakharin** *, **Timothy C. Bates**

Department of Psychology, University of Edinburgh, Edinburgh, Scotland, United Kingdom

* s1775682@sms.ed.ac.uk

**Data Availability Statement:** The data and materials used in this paper as well as R code used to generate the results are openly available at the OSF site for this paper at https://osf.io/3ypxu/, DOI 10.17605/OSF.IO/3YPXU.

## Abstract

Relational models theory predicts that social relationships are formed from four underlying psychological models: communal sharing, authority ranking, equality matching, and market pricing. Here, in four studies, we test this four-factor model using the 33-item Modes of Relationships Questionnaire (MORQ). In Study 1, we administered the MORQ to N = 347 subjects. A parallel analysis supported the four-factor structure, but several items failed to load on their predicted target factors. In Study 2 (N = 617), we developed a well-fitting four-factor model of the MORQ with a total of 20 items (five items retained for each factor). This model replicated across multiple relationships reported by each subject. In Study 3, we replicated the model in an independent dataset (N = 615). A general factor associated with relationship type was required in both Study 2 and Study 3. In Study 4, we tested the nature of this general factor, finding that it was associated with the closeness of the relationship. The results support the Relational Models four-factor structure of social relationships. Given the mature theory and applications in a wide range of disciplines, from social to organisational psychology, we hope that this compact, valid, and interpretable instrument leads to increased usage of the scale.

## Introduction

Relational models theory (RMT) offers a comprehensive model of interpersonal relationships [1, 2]. The theory proposes that relationships are represented and processed within four underlying psychological models: Communal Sharing, Equality Matching, Authority Ranking and Market Pricing. A measure of these models–the Modes of Relationships Questionnaire (MORQ) [3]–has been developed, permitting testing of the theory. Analyses of the MORQ, however, have found a poor fit to the theorised four-factor model [3–5]. In the present paper, we set out to locate the cause of this poor fit, generate a well-fitting model, and establish the replicability of the newly proposed model. Below, we briefly introduce the RMT and the questionnaire associated with it.

Based on ethnographic fieldwork and a review of previous studies, Fiske [1] proposed four distinct relational systems constituting the structures of social relationships. These four models are theorised as fundamental and innate and serve as a comprehensive framework to describe all possible human relationships [6]. They depict how individuals evaluate their status in relation to others and elucidate appropriate or inappropriate behaviours in a given

**Funding:** The author(s) received no specific funding for this work.

**Competing interests:** The authors have declared that no competing interests exist.

social context. In essence, they offer a framework for comprehending social interactions and the expected norms of behaviour in diverse social settings. The first of these models, *Communal Sharing* (CS), focuses on what people have in common and is exemplified in relationships where people share an identity with others, such as family, tribe, religion, or ethnic group, resulting in mutual recognition of social equivalence of individuals. This shared identity is reflected in helping others regardless of their past contributions, treating the property as communal, and making joint consensus-based decisions. The second relational model is *Equality Matching* (EM), in which individuals treat each other as distinct but equal partners. In EM relationships, work inputs and outputs are divided equally where possible. Where resources and work are not divisible equally, individuals keep a count of what they give and receive and equalise this over time. Examples of this relationship include mutual credit organisations and babysitting co-ops. EM also extends to vengeful behaviour, such as eye-for-an-eye justice [7]. The third model, *Authority Ranking* (AR), implements a hierarchy system in which social interactions are based on recognising and respecting different levels of authority. The distribution of resources in this model is expected to be unequal, with superiors feeling entitled to a larger share of resources and subordinates accepting this division as fair [8]. A range of factors can influence ranking in an AR, including age, gender, seniority, and achievement. One example of this model would be the relationship between employer and employee. The fourth and final relational model in RMT is *Market Pricing* (MP). The MP model suggests that people relate to each other based on the value they exchange in a relationship as if it were a market transaction. According to this model, individuals perceive their relationships as a means to obtain desired resources, assistance, or support from the other person. Examples of relationships that align with this model include commercial partnerships, where transactions are prominent, as well as cultural constructs like the concepts of price, wages, or dividends.

While these four models are conceptualised as distinct, RMT predicts that a given human relationship typically reflects combinations of two or more relational models. For example, relationships within a family context usually emphasise the CS relationship. However, children within a family may also be expected to respect their parents (AR relationship), do their fair share of chores (EM relationship) and, perhaps, to be paid for doing some of them (MP relationship).

A substantial amount of empirical research has demonstrated that relational models can accurately predict significant outcomes. For instance, Vodosek [9] found that horizontal collectivism was associated with equality matching and communal sharing relationships, whereas vertical individualism was related to a preference for authority ranking, and vertical collectivism was related to a preference for authority ranking and communal sharing. Biber et al. [10] investigated the relationship between relational models and universal human values [11]. They found that individuals who prioritise CS relationships place greater importance on benevolence and universalism values while placing less emphasis on power and achievement. Conversely, those who value AR or MP relationships tend to prioritise power and achievement values over benevolence and universalism. A disparity between anticipated and real relationship models resulted in a sense of inequity among employees at work [12, 13], and they began to view their supervisors as lacking morals [8]. In clinical samples, different diagnoses were linked to either difficulties or extreme use of specific relational models [14, 15]. For instance, dysthymia was found to be positively associated with high levels of AR relationships with close friends and family members, while hypomania was positively associated with high levels of CS and EM relationships with authority figures.

## Updates and applications of RMT

Other models of social relations have been developed both before and since the innovations of RMT. Perhaps the key feature distinguishing RMT from theories of social relationships, such as interdependence theory [16], attachment theory [17] and social identity theory [18] is the emphasis RMT places on explaining the underlying structure of relationships. Rather than focusing on the role of interdependence within relationships, emotional bonds formed early in life or the sense of self derived from a social group membership, RMT provides a framework for understanding social interactions and the appropriate behaviours within them.

RMT has continued to evolve and expand its realm of application, with several changes being of particular relevance. First, a personality assessment tool–the Relationship Profile Scale [15], was developed to evaluate individual preferences for distinct relational models, measuring the perceived importance, satisfaction, challenges, and motivations associated with each of the four relational models. Together with the MORQ, the Relationship Profile Scale enables a comparison of individuals' desired and actual relationship experiences. A significant theoretical advance known as Relationship Regulation Theory [19] extended RMT into the domain of moral psychology by associating each relational model with four distinct moral motives. For instance, the moral motive of hierarchy is based on the AR relationship and its focus on establishing and upholding a clear ranking in social groups. The motive of hierarchy motivates those in lower positions to show respect, obedience, and deference to those above them, including leaders, ancestors, or gods, and to punish those who go against them. Conversely, those in higher positions feel a moral responsibility to guide, direct, and safeguard those below them. This expansion links RMT to existing moral theories [20, 21] but construes the nature of moral behaviour as relationship management and emphasises that the moral value of acts such as harming, unequal treatment, or being impure are dependent on the relationships and relational models within which they are deployed.

Most recently, RMT has undergone another significant enhancement by incorporating the well-established effects of incentives on behaviour into our understanding of relationships and relationship management. Known as Relational Incentives Theory [22], this extension posits that for incentives to be effective, they should align with relational models. For instance, incentives promoting communal sharing relations should be most effective when they align with the motive of unity, while proportional incentive schemes work best for market pricing relations. These recent advancements demonstrate the continued significance of RMT and highlight the crucial role of the four relational models in comprehending and predicting diverse behaviours, ranging from resolving moral disagreements to determining the efficacy of incentive schemes.

## Measuring relational models

Realising the benefits of an instrument to test the predictions of RMT, Haslam and Fiske [3] developed the Modes of Relationships Questionnaire (MORQ), a 33-item instrument to measure the four social relationships specified in RMT. For each relationship, items were constructed to tap into each of eight classes of behaviour predicted to be influenced by social relationships: 1) distribution and use of resources, 2) work, 3) morals, 4) exchange, 5) decision-making, 6) social influence, and 7) identity, with an eighth "miscellaneous" category reserved for behaviours specific to each particular relationship mode. The EM relationship has two items in this miscellaneous category.

The MORQ has a slightly unusual administration process. Participants first generate a list of relationships they have with others, typically 40, from which 10 are selected randomly to avoid oversampling easier-to-recall relationships. They then rate each relationship on each of the 33 MORQ items. This creates data in which information from each participant generates

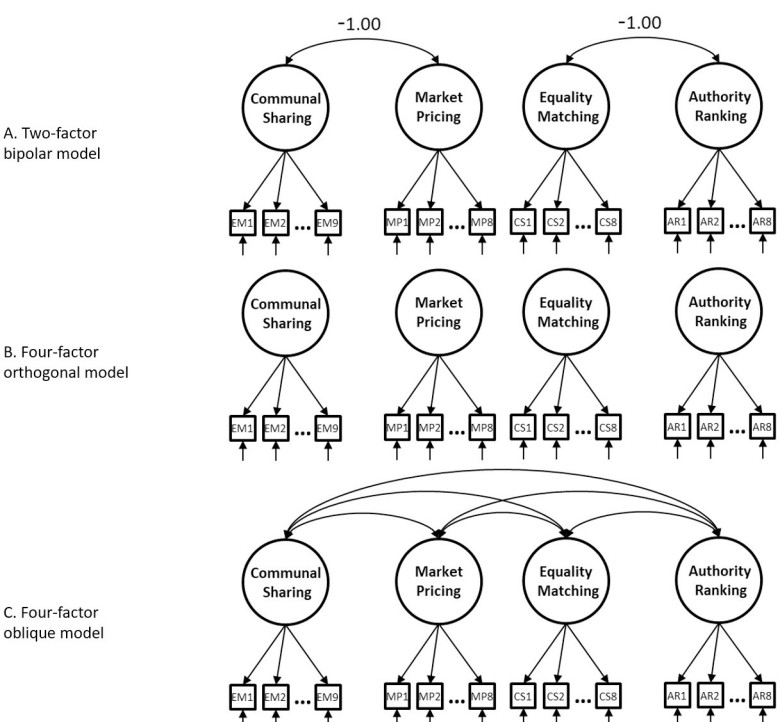

**Fig 1. Three models tested by Haslam and Fiske [3].** a) 2-factor Bipolar Model, b) 4-factor Orthogonal Model and c) 4-factor Oblique Model.

correlated information on multiple target individuals. Between-participant variance is statistically removed to compensate for the dependency amongst responses so that the data analysed consisted of each participant's deviation from their own mean rating across all items.

During the initial study, Haslam and Fiske [3] administered the questionnaire to 42 participants. Three theorised models were compared (see Fig 1): a) A two-factor model consisting of orthogonal bipolar dimensions, one running from EM to AR and one from CS to MP, thus capturing the equality-inequality and closeness-distance dimensions; b) A four-factor orthogonal and c) a four-factor oblique model. Confirmatory factor analyses preferred the four oblique factor model over the two other models [3]. However, the absolute fit of this four-factor model was well below the accepted criteria (GFI = .75, RMSEA = .243).

Since this initial report, only a few studies have assessed the psychometric properties of the MORQ. Brito and colleagues [4] evaluated the four-factor structure of the MORQ in a sample of 63 Portuguese participants confirming that the fit metrics of the model were unsatisfactory (GFI = .74, RMSEA = .092). Vodosek [9] selected a different 22-item set from the 33-item MORQ, administered these to a US sample (N = 465), and tested the fit of a four-factor model. However, even this reduced set of items did not fit a four-factor model well (CFI = .80, GFI = .83, RMSEA = .07). In this study, however, participants were asked to indicate the degree to which they believed each MORQ statement should be true in an ideal working group, rather than rating their actual relationships. This difference in approach may account for the low level of fit observed. Finally, Bogodistov and Lizneva (5) used the MORQ with a Ukrainian sample of 99 metallurgical workers. They modelled three of the four scales, excluding EM (based on poor reliability) and also excluded 11 items from the remaining three scales based on low factor loadings, leaving a total of six items on the AR scale, four items on the CS scale

and three items on the MP scale. A structural model of these items representing only 13 items and three factors showed a good model fit (CFI = .963, RMSEA = .058).

These previous attempts to model the MORQ suggest that some items have low validity (based on low factor loadings). They were also hampered by small samples, lacking the power to detect item structure reliably. Given the lack of fit for the simple four-factor models tested, it may also be that more complex structural models are needed to account for variance in the MORQ. Previous research, therefore, cannot be interpreted as rejecting the RMT but rather suggests the need for additional research and modelling. To advance the literature, we conducted four studies addressing these shortcomings.

## Study 1a

In Study 1a, we used structural equation modelling to test the fit of the four-factor model (see Fig 1C) proposed by Haslam and Fiske [3] in a large sample. Confirming that the model fits poorly, we then attempted to improve the model in a smaller set of items, retained based on high factor loadings suggested by factor analysis.

### Method

**Participants.** A total of 347 United Kingdom residents (228 women, 118 men, 1 other; mean age 33.96 years, SD = 13.44) were recruited using Prolific Academic, an online research-recruiting system. The data were collected in April-May 2021. The study was approved by the University of Edinburgh PPLS Research Ethics Committee.

**Measures.** Relational models were assessed using the MORQ [3]. This instrument assesses the CS model (eight items, e.g., "*If either of you needs something, the other gives it without expecting anything in return*"); EM (nine items, e.g., "*If you have work to do, you usually split it evenly*"); AR (eight items, e.g., "*One of you is entitled to more than the other*"); and MP (eight items, e.g., "*What you get from this person is directly proportional to how much you give them*").

**Procedure.** Testing was done using the Qualtrics online survey platform. Before starting the study, participants received an explanation of the study and were asked to provide written consent by signing a consent form. After giving informed consent, each participant was asked to identify 40 people with whom they interacted at any closeness level, regardless of how superficially or infrequently, giving a memorable name for each. An automated branching logic in the questionnaire randomly selected one of these names, and the subject was prompted to complete the MORQ, rating this selected relationship. In order to increase the sample size and, consequently, the reliability of the study, each participant was asked to identify 40 relationships but to rate only one of these relationships instead of ten as in the original study [3]. Total testing took approximately 9 minutes per participant on average. All data were de-identified and collected using Prolific IDs to protect participants' privacy. No personally identifying information was collected and the authors did not have access to information that could identify individual participants during or after data collection. For privacy, Prolific IDs have been anonymised and replaced with numerical IDs in the open data associated with this manuscript.

Model fit was assessed using the Comparative Fit Index (CFI), Tucker-Lewis Index (TLI), and the root mean square error of approximation (RMSEA). The RMSEA evaluates the deviation of a hypothesised model from an ideal one. It ranges between 0 and 1, with values closer to zero indicating a better fit. In contrast, the CFI and TLI compare the fit of a hypothesised model to that of a baseline model, which assumes no correlation between any underlying continuous variables. Higher values, closer to 1.0, indicate a better fit for CFI and TLI. Following Hu & Bentler [23] and Yu [24], we adopted criteria of TLI and CFI > = .95 and RMSEA < =

.06. The comparative fit of the models was assessed by the Akaike Information Criterion (AIC) [25], which penalises un-parsimonious models. All statistical analyses were completed in R [26] and umx [27].

## Results

Descriptive statistics and Cronbach's alpha coefficients for the four relational models are given in Table 1. Cronbach's alphas ranged from 0.72 to 0.86 suggesting good internal consistency of the four scales.

We first tested the best-fitting model of the MORQ presented by Haslam and Fiske [3], consisting of four factors with items loading only on their corresponding factor and the factors themselves permitted to correlate. This model had unsatisfactory fit, $\chi^2$ (489) = 1563.59, p < 0.001; CFI = 0.738; TLI = 0.717; RMSEA = 0.08.

To explore the cause of this lack of fit, we conducted a parallel analysis [28] followed by an exploratory factor analysis using a promax (oblique) rotation. The parallel analysis supported a four-factor structure, with the first four factors accounting for 11.3%, 9.8%, 9.1% and 7.7% of the variance in MORQ scores, respectively. The exploratory factor analysis extracting four factors indicated some likely problems. Eight items had a larger loading on a factor other than that they intended to assess. Ten items had cross-loadings over .30, suggesting that they measured more than just one relational model.

Based on this factor analytic evidence and on previous studies indicating that some items in the MORQ loaded poorly on their target factor [5, 9], we attempted to create an abbreviated 12-item scale (three items per factor to identify the model). Our selection criteria were high (> .50) loadings on their target factor and low off-factor loadings (< .20). Twelve items meeting these criteria were found which supported a well-fitting model, albeit in the same data set in which they had been discovered ($\chi^2$ (48) = 107.53, p < 0.001; CFI = 0.943; TLI = 0.922; RMSEA = 0.06).

## Study 1b

As the analyses of Study 1a were exploratory and therefore prone to yield unreplicable results [29], we attempted to replicate the final model in an independent dataset. In order to permit control of between-participant variance, we also asked participants in this new sample to rate ten individuals as in Haslam and Fiske [3], rather than just a single target individual, as we had done in Study 1a.

### Method

**Participants.**  A total of 135 United Kingdom residents (100 women, 33 men, 2 other; mean age 35.74 years, SD = 14.27) were recruited using Prolific Academic, an online research-recruiting system.

The data were collected in May 2021. The study was approved by the University of Edinburgh PPLS Research Ethics Committee.

**Table 1. Descriptive statistics for Study 1a variables.**

| Relational mode | M | SD | α |
|---|---|---|---|
| Communal Sharing (CS) | 3.92 | 1.23 | 0.82 |
| Equality Matching (EM) | 4.25 | 1.1 | 0.78 |
| Authority Ranking (RM) | 3.17 | 1.37 | 0.86 |
| Market Pricing (MP) | 3.59 | 1.05 | 0.72 |

**Measures and procedure.** Relationships were assessed using 12 MORQ items selected in Study 1a. Testing was done using Qualtrics online survey platform. Before starting the study, participants received an explanation of the study and were asked to provide written consent by signing a consent form. As in Study 1a, each participant identified 40 people with whom they interacted at any level, giving a memorable name to each. An automated branching logic in the questionnaire then randomly selected ten of these names. The subject was then prompted to complete the 12-item version of the MORQ, rating each of the selected relationships. Total testing took approximately 13 minutes per participant on average. All data were de-identified and collected using anonymous codes to protect participants' privacy. No personal identifying information was collected, and the authors did not have access to any information that could identify individual participants during or after data collection.

## Results

Before conducting inferential analyses, following Haslam and Fiske [3], the impact of reporter-specific variance in the multiple target reports from each subject was controlled. Where Haslam and Fiske accomplished this by dummy coding the participant IDs and residualising the data for these dummy variables, we accomplished the same purpose in a multi-level analysis, with participant ID as a random variable, again retaining the unstandardised residuals.

We assessed the fit of the four correlated factor 12-item model developed in Study 1a. Unfortunately, the model fit poorly in this new sample ($\chi^2$ (48) = 520.73, p < 0.001; CFI = 0.902; TLI = 0.865; RMSEA = 0.086), indicating a failure of replication.

## Discussion of Study 1a and 1b

The aim of studies 1a and 1b was to test if a well-fitting model of the MORQ was possible and if this reliably supported the RMT. While a factor analysis supported evidence for four factors in the MORQ, it also showed that a substantial number of items either failed to load on their corresponding factor or showed large cross-loadings on other factors. While we could identify 12 items from this analysis such that three items were available for each predicted relationship model and fitted a 4-factor model, this model failed to replicate in an independent sample. Two possible accounts for this present themselves. First, the theoretical four-factor structure may be valid, but perhaps because of a small discovery sample, we were unable to select items which reliably assess this true structure, and instead, our item selection capitalised on sample-specific variance. Alternatively, the model replication may have failed because the four-factor structure itself is incorrect or incomplete. For example, it may be necessary to replace correlations between factors with a general relationship factor, representing a general tendency to initiate or avoid relationships with other people or to make some other model modifications. To address these possibilities, we conducted a second study with a larger discovery sample, tested a wider range of models in this sample, and requested five rather than one relationship from each participant, allowing us to validate the models across a range of participant responses.

## Study 2: Alternate models and larger sample

Study 1 failed to find the well-fitting replicable structure of the MORQ. Although factor analysis indicated that four factors are needed to explain the variance in the scale, several items failed to load on the expected factors. Post-hoc 12-item model based on items that factor analysis suggested should be retained as relatively pure indicators of each of the four domains also failed to replicate. To address the Study 1 problems, in Study 2, we collected a larger sample

and used a multi-trait multi-method approach to develop a well-fitting model of the MORQ and to test if this new model replicates well.

## Method

**Participants.**   A total of 617 people (309 women, 304 men, 4 other; mean age 39.00, SD = 14.39) from the United Kingdom were recruited using Prolific Academic. The data were collected in January-February 2022. The study was approved by the University of Edinburgh PPLS Research Ethics Committee.

**Measures and procedure.**   Participants' endorsement of relational models was measured using the full 33-item Modes of Relationships Questionnaire (MORQ) [3]. The questionnaire was hosted on the Qualtrics survey platform. Before starting the study, participants received an explanation of the study and were asked to provide written consent by signing a consent form. After providing informed consent, each participant generated a list of 40 relationships. Qualtrics automation was then used to select five relationships at random, and for each of these, the subject was asked to complete the online MORQ with respect to this relationship. Total testing took approximately 19 minutes per participant on average. All data were de-identified and collected using anonymous codes to protect participants' privacy. No personal identifying information was collected, and the authors did not have access to any information that could identify individual participants during or after data collection.

## Results

Descriptive statistics and Cronbach's alpha coefficients for the four relational models are given in Table 2. Cronbach's alphas ranged from 0.74 to 0.87 suggesting good internal consistency of the four scales.

To generate our model of the MORQ, we used only the first relationship (out of five reported by the participants), treating the remaining four relationships as internal hold-out replication datasets. Our initial model used a four-factor intercorrelated structure. To explore which, if any, sets of items would permit fit this structural model, we used a function designed to select the best items while keeping the factor structure intact. Procedurally, the function removed items one by one, starting from those that fit the model least well. This item removal process continued until the model reached a satisfactory model fit by at least two out of three criteria (CFI and TLI $> = .95$; RMSEA $< = .06$) [23]. The function is documented in the OSF site for this paper.

The automatic function removed four items from each of the EM, CS, and MP scales and three items from the AR scale, yielding a model which achieved a good fit in the test dataset but which did not replicate perfectly in the hold-out relationship datasets (see Table 3).

While the drop-off was not substantial, we wished to investigate whether more complex models would reliably yield a good fit. Based on evidence that the four relational models are typically correlated [3, 4], we tested the effect of removing the intercorrelations among the factors and instead modelling item covariance via a general factor loading on all items to the

**Table 2. Descriptive statistics for Study 2 variables.**

| Relational Model | *M* | *SD* | α |
|---|---|---|---|
| Communal Sharing (CS) | 4.15 | 1.33 | 0.85 |
| Equality Matching (EM) | 4.41 | 1.13 | 0.80 |
| Authority Ranking (RM) | 3.25 | 1.44 | 0.87 |
| Market Pricing (MP) | 3.72 | 1.09 | 0.74 |

**Table 3. Testing a model with four intercorrelated factors (model fit for test and four replication datasets in Study 2).**

| Data subset | CFI | TLI | RMSEA |
|---|---|---|---|
| 1st relationship (test) | 0.963 | 0.956 | 0.036 |
| 2nd relationship (replication 1) | 0.916 | 0.901 | 0.05 |
| 3rd relationship (replication 2) | 0.932 | 0.919 | 0.049 |
| 4th relationship (replication 3) | 0.947 | 0.938 | 0.043 |
| 5th relationship (replication 4) | 0.923 | 0.908 | 0.051 |

**Table 4. Model fits for final model with general factor (test and four replication datasets in Study 2).**

| Data subset | CFI | TLI | RMSEA |
|---|---|---|---|
| 1st relationship (test) | 0.956 | 0.944 | 0.044 |
| 2nd relationship (replication 1) | 0.939 | 0.923 | 0.047 |
| 3rd relationship (replication 2) | 0.962 | 0.951 | 0.039 |
| 4th relationship (replication 3) | 0.955 | 0.943 | 0.044 |
| 5th relationship (replication 4) | 0.965 | 0.955 | 0.038 |

model. Examining the sources of a model misfit in earlier analyses also suggested a unique bivariate link between the CS and EM factors, which was added. Applying the same automated function to this new model supported this modification. A total of four EM and three CS, MP, and AR items were dropped, yielding a model with five indicators of each relational factor and resulting in a well-fitting model ($\chi^2$ (149) = 990.79, p < 0.001; CFI = 0.956; TLI = 0.944; RMSEA = 0.044) which also replicated well across the holdout datasets. The RMSEA remained below the threshold in all tests while other fit indices improved or decreased in proportion as compared to the fit obtained in the initial data (see Table 4). Fig 2 shows the final model (using the dataset with all five relationships combined after controlling for between-subject variance). The final item set used in Study 2 is listed in the S1 Appendix.

## Discussion of Study 2

In study two, we were able to create a well-fitting model of the MORQ retaining five items per relational mode. We also found that the model required a general factor loading on all items, which was not investigated in previous studies relying on factor correlations only. This model performed satisfactorily across holdout data from the same dataset, providing additional support for the validity of the four-factor MORQ model. One limitation of our study, however, is that the same participants generated both the discovery (the first reported relationship) and replication (the second to the fifth reported relationship) data. While the model replicated in the different target data provided by our subjects, we wished to replicate the model in a completely independent dataset to further corroborate the new model. We, therefore, conducted Study 3, testing the exact final model from Study 2 in a new dataset.

## Study 3

Our objective in Study 3 was to validate the 20-item model developed in Study 2 by replicating the results in an independent sample. We made no changes to the model, and the measures were identical to those used in Study 2. We expected to confirm the model structure using the same fit metrics as in Study 2 (TLI, CFI and RMSEA) and refine the model if needed.

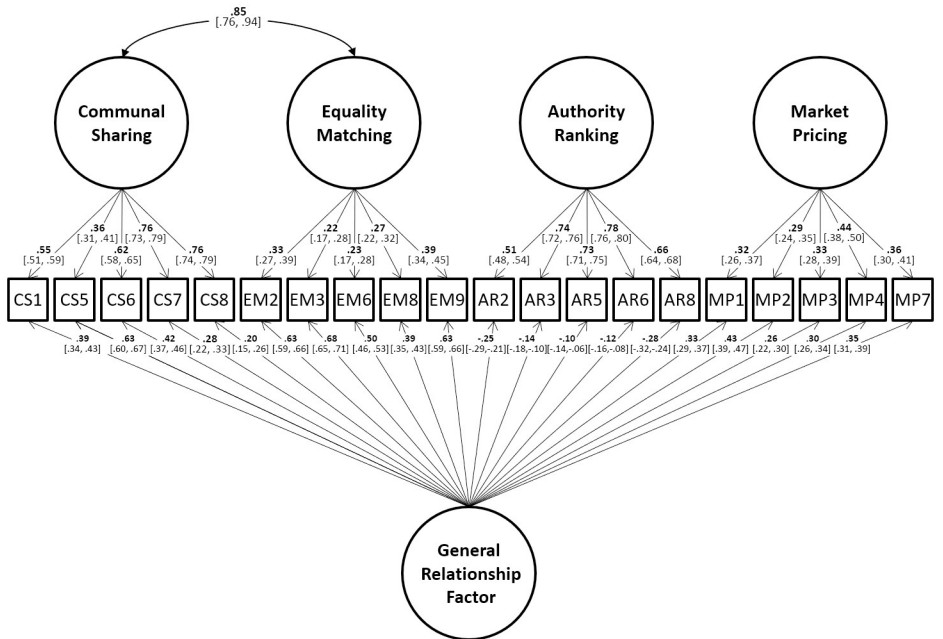

**Fig 2. Structure of the well-fitting MORQ model in Study 2.**

## Method

**Participants.**   A total of 615 people (307 women, 305 men, 3 other; mean age = 41.57, SD = 13.97) from the United Kingdom were recruited using Prolific Academic. Participants from studies 1 and 2 were excluded in order to ensure dataset independence. The data were collected in July 2022. The study was approved by the University of Edinburgh PPLS Research Ethics Committee.

**Measures and procedure.**   In Study 3, we followed the same procedure and used the same materials (20-item MORQ with five items per each relational mode) as in Study 2. Total testing took approximately 15 minutes per participant on average. All data were de-identified and collected using anonymous codes to protect participants' privacy. No personal identifying information was collected, and the authors did not have access to any information that could identify individual participants during or after data collection.

## Results

We fitted the exact model developed in Study 2 to the new dataset collected for Study 3 and examined its fit. The model replicated well showing excellent fit ($\chi^2$ (149) = 990.79, p < 0.001; CFI = 0.955; TLI = 0.943; RMSEA = 0.043). In addition to a good fit, the factor loadings were also comparable to those found in Study 2. Moreover, the correlation between the CS and EM factors was also very similar (.85 vs .80 in Study 2). The replicated model is shown in Fig 3).

Full details of the model are tabulated on the OSF site for this paper.

## Discussion of Study 3

Study 3 successfully replicated the four-factor 20-item model of the MORQ developed in Study 2 in an independent dataset. Despite no structural changes to the model, the model fit metrics were excellent and comparable to those in Study 2. We can have confidence, therefore,

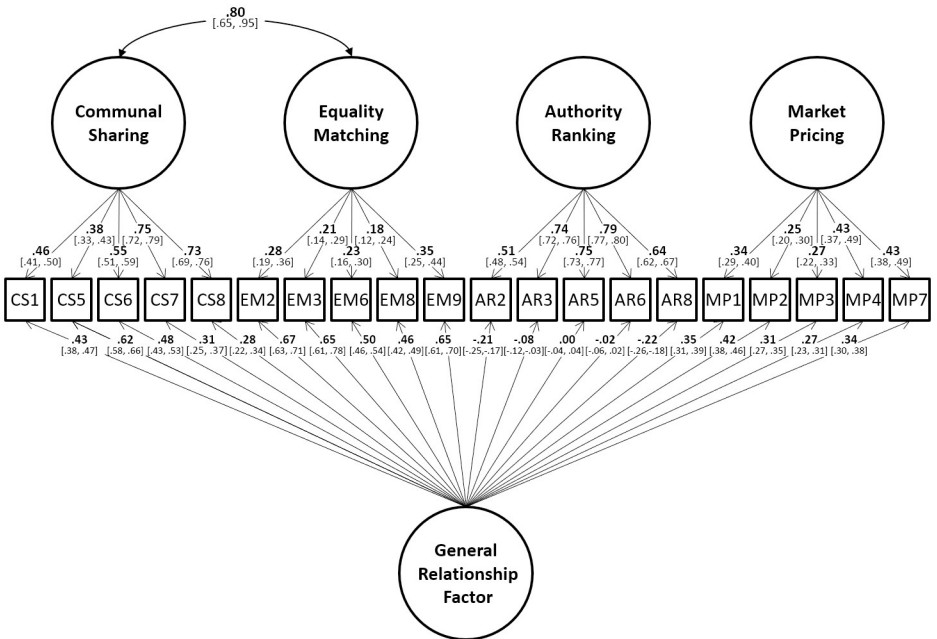

**Fig 3. The structure of the well-fitting MORQ model in the replication data, Study 3.**

that this is a reliable, well-fitting, and useful model of MORQ. Study 3 also confirmed the presence of a general factor in the MORQ, as shown in Study 2. This general factor had high loadings on CS, EM and MP relational model items and low (or negative) on AR relational model items. Given that the general factor was required in two independent datasets, we believe it is not a sampling error artefact and therefore requires further explanation. For instance, this factor may assess general commitment and devotion to form social relationships with others, with low scores representing a "null relationship" [2], an avoidant attitude towards forming relationships. However, traits such as socially desirable responding, differential emphasis on traditional social conventions and authorities across the four types, or artefacts of factors such as the closeness of the relationship to the respondent all could also account for some or all of the variance in the general factor. To explore these possibilities, we re-contacted the participants from studies 2 and 3 and tested associations with the general factor.

## Study 4

In Study 4, we investigate the nature of the general factor that emerged in both Study 2 and Study 3. We tested three possible explanations. First, several MORQ items describe socially desirable behaviours (e.g. "*If either of you needs something, the other gives it without expecting anything in return*"). For this reason, we hypothesised that the general factor might represent social desirability bias, exaggerating desirable traits due to honest self-deception or conscious impression management [30]. To test this hypothesis, we administered the Balanced Inventory of Desirable Responding (BIDR) [31] because it allows testing both types of bias: deliberate impression management and self-deceptive enhancement.

Second, given that in both studies 2 and 3, the general factor correlated negatively with AR items but positively with all items defining the other three MORQ models, we speculated that a simple authoritarian/non-authoritarian distinction could drive the general factor. To test this speculation, we asked participants to fill in the Right Wing Authoritarianism

questionnaire (RWA) [32], which measures authoritarian personality traits such as submission to traditional authorities and social conventions.

Finally, we hypothesised that the general factor might reflect the specific relationship with the individual a respondent was rating. To test this, we asked participants to recall the relationships they reported during Study 2 and Study 3 data collection and to classify each of these individuals by type (e.g. "colleague" or "close family"). We coded this measure as a categorical variable with eight unordered levels. We expected higher general factor loadings for close relationship types (such as close family or close friend) and lower loadings for relationships that are typically less close (e.g. an employer or service personnel).

## Method

**Participants.** Study 2 and Study 3 participants were re-contacted on the Prolific academic platform 3–8 months after data collection from Study 2 and Study 3 was completed. A total of 447 participants agreed to participate in the follow-up study. The data were collected in September-October 2022. The study was approved by the University of Edinburgh PPLS Research Ethics Committee.

**Measures and procedure.** Before starting the study, participants received an explanation of the study and were asked to provide written consent by signing a consent form. After providing informed consent, the following three questionnaires were administered.

**Right Wing Autoritarianism questionnaire (RWA) [32].** The RWA is a 22-item instrument measuring the tendency to defer to authorities, endorsement of traditional values, and support for aggression toward outgroups. Participants respond to a series of statements (e.g., "*What our country really needs is a strong, determined leader who will crush evil, and take us back to our true path*") on a nine-point Likert scale ranging from 1 (Strongly disagree) to 9 (Strongly agree).

**Balanced Inventory of Desirable Responding (BIDR) [31].** The BIDR is a 40-item instrument measuring the tendency to overstate one's socially desirable behaviour and personality traits. The BIDR contains two separate 20-item measures, Impression Management, designed to test conscious self-presentation (e.g. "*I have never dropped litter on the street*"), and Self-Deceptive Enhancement (e.g. "*My first impressions of people usually turn out to be right*"). BIDR is scored on a seven-point Likert scale ranging from 1 (Not true) to 9 (Very true).

**Relationship type.** We asked participants to recall the relationships they reported during the original data collection in studies 2 and 3 and to classify them by type. We coded the reported type of the relationship as a categorical variable with eight levels ("Your manager or employer", "Your employee", "Service personnel", "Acquaintance", "Colleague", "Distant family", "Close friend", "Close family".

The measures were hosted online on the Qualtrics survey platform. Total testing took approximately 8 minutes per participant on average.

## Results

First, we tested the hypothesis that authoritarianism explains the general factor scores extracted from the model using the umx function umxFactorScores(). Regression scoring was used to determine the factor scores, and potential confounding effects of authoritarianism were tested in each of the five relationships examined. The results showed no evidence of any association between the general factor and Right-Wing Authoritarianism (RWA). The correlation between the general factor and RWA was not significant in any of the five relationships, with correlations ranging from -.06 to .06 (e.g., in the relationship 1, r (437) = -.03, p = .504).

**Table 5. Study 4 regression results using the general relationship factor as the criterion and the manager/employer relationship category as a reference group.**

| Relationship category | b | 95% CI |
|---|---|---|
| (Intercept) | -0.76** | [-0.96, -0.56] |
| Service personnel | 0.35* | [0.01, 0.70] |
| Acquaintance | 0.60** | [0.38, 0.81] |
| Distant family | 0.65** | [0.40, 0.90] |
| Close family | 0.71** | [0.49, 0.92] |
| Colleague | 0.78** | [0.57, 1.00] |
| Employee | 0.84** | [0.45, 1.22] |
| Close friend | 1.05** | [0.84, 1.26] |

*Note.* b represents regression weights.

* indicates p < .05.

** indicates p < .01.

Next, we tested whether the general factor was explained by social desirability, specifically self-deceptive enhancement and impression management scales of the BIDR. The results showed that the general factor was unrelated to both self-deceptive enhancement (e.g., in the relationship 1, r(527) = .03, p = .478) and impression management (r(527) = .04, p = .401) scales across all five relationships, contrary to our hypothesis.

Finally, we used regression to test the hypothesis that relationship type (dummy-coded with eight factors as indicated above) explains the general factor. We found that this was significant, explaining 6.2% of the variance (F(7, 2139) = 21.26, p < .001). The beta coefficients for each of the eight types of relationships can be seen in Table 5. Fig 4 shows a boxplot depicting the relationship between the general factor and eight relationship categories.

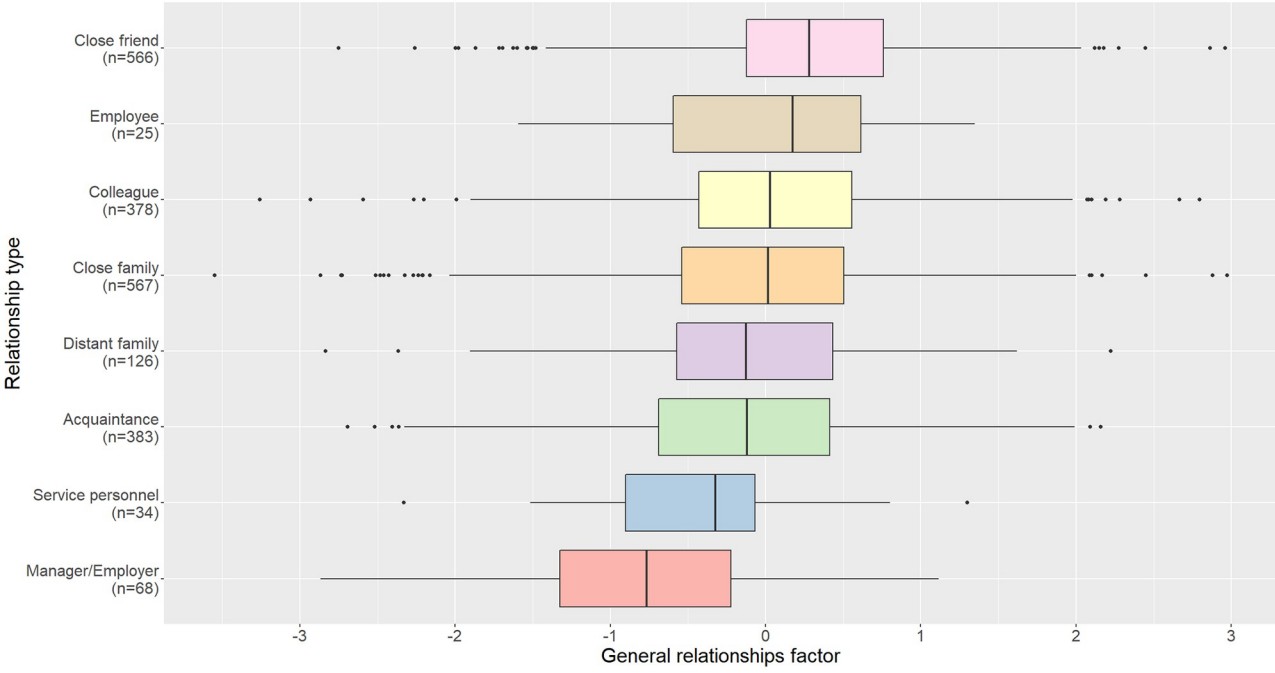

**Fig 4. The relationship between the general relational factor and eight types of relationships used in Study 4.**

## Discussion of Study 4

Study 4 tested three possible causes of the general factor: authoritarianism, social desirability bias, and relationship type (e.g. colleague, close friend, etc.). While RWA (assessing authority) and BIDR measures (measuring socially desirable responding) were unrelated to general factor scores, relationship type did account for a portion of the variance in the general factor. High general factor loadings were associated with closer relationships (e.g. 'close friend'). In contrast, low scores were associated with relationship types that are usually less close (e.g. 'manager/employer' or 'service personnel'), suggesting that the general factor may represent the relationship's closeness. The employee relationship had a much higher score on the general factor than the employer relationship, despite both representing the AR relationship. We believe this may reflect the paternalistic side of authoritarian leadership in which an employer's role involves, to a degree at least, responsibility towards their employees. We discuss these findings in more depth in the general discussion.

## General discussion

This paper aimed to test whether the MORQ measures four relationship models, as predicted by the RMT, and whether it accurately identifies the proposed structure of social relationship taxonomy. In the three studies reported above, we confirmed the existence of the original four factors, along with support for a general relationship factor. This new model of the MORQ has several implications for RMT and suggests additional directions for research. Each of these is discussed below.

Our main results (studies 2 and 3) supported the Haslam and Fiske [3] four-factor model of social relationships. The model demonstrated a good fit after eliminating items with significant cross-loadings and items that loaded on factors other than their intended ones. This refinement resulted in a model consisting of five items for each social relationship mode. The model also required some minor structural changes. Instead of four intercorrelated factors, the model required a general factor at the item level–in some ways, a more interpretable structure than the six factor intercorrelations it replaced. The general factor loaded positively on CS, EM and MP items but negatively on AR items.

We also found that EM and CS factors are highly correlated (r = .85 in Study 2 sample and r = .80 in Study 3 sample). This is consistent with the original Haslam and Fiske [3] findings, where these factors were positively correlated (r = .60). Despite the high correlation, combining these factors into one relationship worsened the model's fit in our data, suggesting that these two relational models are distinct, but usually work together to define actual relationships. Our model was successfully replicated in an independent dataset, providing further support for the Relational Models four-factor structure of social relationships.

In Study 4, we tested the possible meaning of this general factor by undertaking three additional tests. First, we speculated that the general factor might reflect social desirability, overreporting desirable traits due to cognitive bias or conscious impression management. We tested this explanation by including two measures of social desirability [31], impression management, measuring conscious attempt to enhance self-presentation to others and self-deceptive enhancement, measuring an honest overestimation of one's positive traits. Both these measures were not related to the general factor. This indicates that social desirability bias is not a major concern for the MORQ questionnaire.

As a second possible explanation, we tested whether the general factor measures a broad tendency to construe relationships based on a hierarchy. We theorised that CS, EM, and MP relational models describe relationships of individuals with the same status, whereas AR model explicitly implies unequal status. This would predict a significant negative relationship

between the general factor and authoritarianism, but this was not the case; in our data, the correlation between the general factor and right-wing authoritarianism was not significant. Thus, we feel comfortable concluding that the general relational factor does not reflect hierarchical tendencies.

Finally, we tested whether the general factor represents the closeness of the relationship. We hypothesised that close relationships (those between close friends and family) should manifest as higher in CS and EM relational models than AR and MP, as is reflected in the general factor loadings. Our measure of reported relationship type correlated significantly and positively with the general factor. However, the strength of this relationship was weak, suggesting that relationship closeness is only a partial explanation of the general relational factor. As higher scores on this factor typically occurred for closer relationships, a useful direction for future work would be to study a tendency to invest in building connections. This would be consistent with the concept Fiske [2] termed general commitment and devotion to form social relationships with others, with low scores representing a "null relationship", an avoidant attitude towards forming relationships.

Although our manuscript primarily aimed to enhance the MORQ's psychometric properties and confirm its four-factor structure, our findings also provide insights into the underlying structure of social relationships. Our results suggest that relationships are structured around, at a minimum, these four models of interpersonal relations. Moreover, our findings refute the notion that these four models are merely consequences of a simpler, two-dimensional model (such as equality-inequality or close-distant) since these models did not adequately fit our data.

## Limitations and future directions

We should keep in mind the limitations of the study. The present study supported a self-report measure of relational models with five instead of eight to nine items measuring each relational model. As the original 33 items were designed to cover a spectrum of behavioural domains, generating new items to replace the missing items may be of value to capture the complete spectrum of relationship models. That said, the scales developed here and scored by averaging responses for each scale should be valid for their intended purposes or for identifying relationship models. Of course, a further limitation is that we cannot rule out that other relational models may exist–seeking evidence for relationships that do not fit the four-model structure would be informative regarding the validity and generality of the broader theory. The present studies also were conducted thirty years after the initial study and in a different yet related culture (the UK compared to the US). This may partially account for the finding that some original items did not load on the expected factors. The finding that despite the three decades having elapsed and testing in a much changed and different culture in the UK, the model was validated is a testimony to the durability of the RMT model. However, international, cross-cultural replication of the model in non-western samples and further examination of the nature of the general relational factor are required. The future directions for this more compact, valid, and highly interpretable instrument appear wide. The four social models identified in this study represent universal building blocks of relationships and can be applied across a range of psychological sub-disciplines to gain insights into specific domains of social interactions. For example, within the context of parent-child relationships, one may explore how the four models manifest in parent-child interactions and their impact on child development. Similarly, within the field of education, the four models may be used to examine the dynamics of teacher-student relationships and their impact on student achievement and well-being. By mapping these universal models onto discipline-specific structures, researchers can gain a deeper understanding of the role of social interactions in various contexts and develop tailored

interventions to promote positive relationship outcomes. For instance, workplace and organisational psychology is a particularly suitable discipline for the application of reliable, valid measures that can diagnose the current disposition of relationships among staff across different levels of business units or larger structures. These measures can test the alignment of these models with the intended and desired business strategy and assess the efficacy of interventions designed to incentivise relationship change where necessary. Surveying organisations using these measures can reveal if relations designed to primarily embody hierarchy and proportionality are functioning as intended. Additionally, these measures can test the association of incentives with the strength of reported models in a given relationship or modulate incentives to assess the effects predicted by the Relational Incentives Theory. Thus, these measures can add significant value to organisational research and practice.

## Conclusion

Our study aimed to establish the validity and psychometric structure of the MORQ, resulting in a compact, reliable, and valid instrument suitable for use in various applied settings. By providing an efficient means of measuring individuals' preferences for different relational models, the validated scale can be used to further explore and apply RMT. Our findings suggest that the MORQ can be a useful tool in both research and applied settings, facilitating a deeper understanding of the role of relational models in human behaviour and well-being.

## Supporting information

**S1 Appendix. Items retained in final relational models scale, validated in Studies 2 and 3.**
(DOCX)

## Author Contributions

**Conceptualization:** Michael Zakharin, Timothy C. Bates.

**Data curation:** Timothy C. Bates.

**Formal analysis:** Michael Zakharin.

**Project administration:** Timothy C. Bates.

**Software:** Michael Zakharin.

**Supervision:** Timothy C. Bates.

**Writing – original draft:** Michael Zakharin.

**Writing – review & editing:** Michael Zakharin, Timothy C. Bates.

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
