## [Decision Letter · Decision Letter 0]

3 Apr 2023

PONE-D-22-34162Relational Models Theory: Validation and replication for four fundamental relationships.PLOS ONE

Dear Dr. Zakharin,

Thank you for submitting your manuscript to PLOS ONE. After careful consideration, we feel that it has merit but does not fully meet PLOS ONE’s publication criteria as it currently stands. Therefore, we invite you to submit a revised version of the manuscript that addresses the points raised during the review process.

Your manuscript requires minor revisions and its acceptance is conditional on addressing all comments - which are rather minor- raised by all the reviewers. Please note that you do not have to reference all the papers suggested by reviewer 3.

We look forward to receiving your revised manuscript.

Kind regards,

Srebrenka Letina, Ph.D.

Academic Editor

PLOS ONE

Journal Requirements:

2. Please change "female” or "male" to "woman” or "man" as appropriate, when used as a noun (see for instance https://apastyle.apa.org/style-grammar-guidelines/bias-free-language/gender).

3. Please provide additional details regarding ethical approval in the body of your manuscript. In the Methods section, please ensure that you have specified the name of the IRB/ethics committee that approved your study.

4. Please provide additional details regarding participant consent. In the ethics statement in the Methods and online submission information, please ensure that you have specified what type you obtained (for instance, written or verbal, and if verbal, how it was documented and witnessed). If your study included minors, state whether you obtained consent from parents or guardians. If the need for consent was waived by the ethics committee, please include this information.

Additional Editor Comments:

The manuscript requires only minor revisions (for details, see comments of reviewers).

Reviewers' comments:

Reviewer's Responses to Questions

**Comments to the Author**

1. Is the manuscript technically sound, and do the data support the conclusions?

Reviewer #1: Yes

Reviewer #2: Yes

Reviewer #3: Yes

2. Has the statistical analysis been performed appropriately and rigorously? 

Reviewer #1: Yes

Reviewer #2: Yes

Reviewer #3: I Don't Know

3. Have the authors made all data underlying the findings in their manuscript fully available?

Reviewer #1: Yes

Reviewer #2: Yes

Reviewer #3: Yes

4. Is the manuscript presented in an intelligible fashion and written in standard English?

Reviewer #1: Yes

Reviewer #2: Yes

Reviewer #3: Yes

5. Review Comments to the Author

Reviewer #1: This manuscript presents five studies that attempt to clarify the structure of interpersonal relationships as assessed by a self-report measure based on relational models theory (RMT). The studies compellingly show that although there is good evidence of four relationship factors, consistent with RMT, the existing measure of those factor is structurally weak. More importantly, they show that the measure can be substantially improved by (a) removing poor items, (b) adding a nonspecific factor capturing relationship closeness (or something similar), and (c) allowing one correlation between two factors (in place of the six included in earlier structural models).

Collectively the studies make several worthwhile contributions. They 1) produce a substantially improved measurement scale that will be useful for researchers interested in RMT; 2) affirm its four-dimensional structure; and 3) provide a helpful and simplifying theoretical account of why model fit of the original scale was poor, namely that it failed to consider generic closeness of relationships separate from their underlying model. In essence, we can understand and measure relationships in terms of the model or models on which they are based, AND on their level of closeness or engagement.

Methodologically the studies are strong. Samples are consistently larger than prior research, structural analyses are carried out carefully and with sophistication, and replication with independent samples is consistently sought and demonstrated.

In addition to these positive comments I have a few small concerns and comments.

1. On line 35, where it says "Relational models theory (RMT) offers a comprehensive classification of interpersonal relationships" it might be better to replace "classification" with "model" or something to avoid the mistaken impression that any relationship must belong to a single model.

2. On line 89, where it says "A two-factor model consisting of orthogonal bipolar dimensions, one running from AR to EM and one from CS to MP", it could be noted what these dimensions mean (e.g., equal-unequal and close-distant). The context here is that whereas RMT proposed four factors, two-dimensional structures like this are popular in social psychology.

3. The Introduction focuses on the issue of how to measure the RMs, without giving even a brief account of the evidence that they matter, such as referring to empirical work showing they predict things that matter. I don't think a full literature review is needed here, but perhaps some citation or work showing the value of the RMT approach would help justify the effort to improve their measurement.

4. A related issue is that the manuscript is present mainly as being about improving a self-report measure and determining its structure, but isn't it also saying something about the actual structure of relationships? Maybe the authors could add a brief statement about whether they think their findings tell us something about how relationships are structured, or do they only tell us about one questionnaire?

5. On line 218 it says "it may be necessary to replace correlations between factors with a general relationship factor". A brief comment about what such a "general factor" is and what finding evidence of it would mean might help here.

6. Line 267 says "which did not replicate well in the hold-out relationship datasets". This comes across as a slight exaggeration: in the replications the CFI is only worse by 3.4% on average and the TLI by 4.1%. Yes there is a drop off, but this is something less than a complete failure to replicate.

7. On line 284, where it says "Fig 2 shows the final model (using the dataset with all five relationships combined)", it's unclear what the combined dataset means. Are the five relationships for each individual treated as independent or is between-subjects variance controlled for? I felt this dataset wasn't explained adequately.

8. On line 297, where it says "We also found that the model required a general factor loading on all items, which was not found in previous studies", it should also be noted that it wasn't found because it wasn't looked for. Past confirmatory studies couldn't have found this factor because they were specifying four RMT factors only.

9. On line 344, "AM" should be "AR".

10. On line 407 it looks very odd to have three correlations in a row that are precisely 0.00.

11. On line 421 and figure 4 it is very interesting that "employee" is much higher on the factor than "employer/manager". This suggests the factor is not about CS/EM/MP versus the AR model in general, as the factor loadings might seem to suggest, but that it's the person's position within the authority relationship that matters. Unless I am misunderstanding it, it seems employers report feel more close to or engaged with their employees than vice versa. This point might merit a brief exploration in the Discussion, as it helps to make the point that the general factor - though the study doesn't entirely nail down what it is - is not just about authority or inequality versus its absence.

Reviewer #2: This paper reports a research program aiming to replicate and improve the MORQ, a measure of the relational models proposed by RMT. The paper is cogent and clearly written. The series of studies is well conceived, and the results are valuable.

The first two studies (1a, 1b) report basically a failed smaller scale effort, before Studies 2 and 3 provide a better and stable solution with more data. Study 4 explores correlation of observed variables with additional measures. One might wonder whether #1 should then be still reported, but I agree that it is good to report it to see the full development. The only thing I would suggest in addition is to test the final model again on the data from Study 1 to provide another internal replication.

It is very commendable that all data from this study are available. Both this paper and the data will be an excellent resource for future work on the MORQ.

One of the many merits of these studies is that the participants were UK residents, in contrast to the participants in the original studies of Haslam and Fiske. Thus the current findings constitute a cross-cultural replication, albeit in a very closely related culture. This should be mentioned, along with the fact that three decades have elapsed since the original studies. These two differences may partially account for the finding that many of the original items did not load on the expected factors.

On p. 9 the authors write that “following Haslam and Fiske (3) between-participant variance was controlled by regression participant ID on all item responses”. Haslam & Fiske (1999) did this: “In each analysis, one item was regressed on 41 dummy variables collectively representing the 42 participants, and the unstandardized residuals were retained.”. Did the current paper do the same? The sentence above is vague.

The method applied by Haslam & Fiske (1999) seems somewhat outdated. Wouldn’t it be better to run the confirmatory factor analysis on a multilevel model, where random variance due to participants is added in the form of a random factor? Is that available in the R packages used here?

The holdout samples in Study 2 fail to replicate the structure according to the criteria set earlier – CFI and TLI > .95, RMSEA < .06. But I noted in Table 2 that RMSEA actually holds up well, it’s the other two indices that are poor. Can this be interpreted? More generally, it would be helpful to the general reader if the authors wrote a few sentences explain what each of the indices assesses, and what the conventions are for acceptable scores, and good scores.

On p. 19, the Discussion includes the sentence “Although we controlled between-participant variance by regressing participant ID on all item responses, these replication datasets were thus not independent.” – This was surprising, is it a copy-paste mistake? Why would you do that procedure if the data are not analyzed in the same model? If you analyze the datasets separately, this regression shouldn’t do anything, and seems unnecessary. Please clarify.

On p. 16, you say “… that this is the true model of MORQ” – I would object to this way of putting it. You found a model that fits the data well, as well as new data. No model is ever true; a model is simply a model! The only question is whether it becomes useful by approximating the data.

Study 4 matched new data to the dataset from Studies 2 and 3. This seems to contradict the earlier statements that “No personal identifying information was collected”. I assume that you matched participants using their Prolific ID? If so, you should clarify that the researcher team could not identify the participants. Please make it clear how you were able to match. Related: I actually had a look at the Study 2 dataset available on OSF and did not see any ID variable for participants, just wave, gender, and age. That should be added?

p. 19: “the general factor and RWA were unrelated (r(2193)” – I couldn’t figure out how this was calculated. If this is a simple correlation, how did you arrive at a variable representing the general factor? Or was this done in the model, and we just see the estimate from the model? But then how do we get an r? Again, wouldn’t this be better done in a multilevel model? Same question for the remaining analyses in Study 4.

The final model features a general “general relationship” factor, which however doesn’t load (or even loads negatively) on the AR items. In addition, the model fit requires a rather high correlation of the latent CS and EM variables. The discussion already provides a good start at explaining these. It’s not fully satisfying, but that’s ok. My question is however: What are the recommendations for future work with the MORQ (in particular the item selection here)? Should researchers simply average items to scales and proceed as usual? Or would it be better to replicate the observed structure, and then either work with the latent variables, or somehow save the (predicted?) values?

I was a bit surprised and perplexed by how speedily participants completed the tasks in each of the studies. Do we know that all participants conscientiously completed all tasks? Was there any check on that?

On lines 44-45, the authors write that Fiske posited the four relational systems to be the dimensions of social relationships. That is definitely not what Fiske posited; he posited that the relational models are four distinct structures of relationships, and not dimensions. Haslam’s subsequent work, using several taxometric methods, supported that conclusion. Regarding the dimensions vs. categories question: this is a bit hard to ask, because H&F1999 doesn’t really discuss this ether, despite the fact that H1994, using data collected with the same instrument, already shows that people perceive their own relationship as categorical. So I would just invite the authors to ponder this.

Lines 45-46 (see also line 69), the authors write that authority ranking entails “respecting and obeying.” In his 1991 book, his 1992 article, and subsequent publications he specifically wrote that command and obedience are not necessary features of authority ranking. In some authority ranking systems, command and obedience are among the cultural implementations of authority ranking, in other cultural implementations, not. If US News and World Report ranks university A above university B, that does not entail anything about commands or obedience. Buddhists regard the Buddha as the highest ranking being, while for Tibetan Buddhists, the Dali Lama is not far below the Buddha. But neither is entitled by their supreme rank to give orders to anyone.

Line 260 has a typo: “sets of items would permit fit this structural model.”

Line 344, “AM” should be ‘AR.’

I wonder whether the “general factor” assesses commitment and devotion to the relational model, where the low end of the scale approaches what Fiske defines as ‘Null relationships.’

Lines 495-6 the authors mention the value of a measure of the person’s proclivity to form relationships based on the four respective relational models. Such a measure exists, and results from studies using it have been published. In essence, after a participant reports 40 relationships, the participant codes how well each one fits the four relational models, and the researcher simply counts how many of the person’s relationships are reportedly structured according to each of the four respective rms. Here are a couple of references:

Allen, Nicholas B., Nick Haslam, and Assaf Semedar 2005. Relationship patterns associated with dimensions of vulnerability to psychopathology. Cognitive Therapy and Research 29: 733-746.

Caralis, Dionyssios, and Nick Haslam 2004. "Relational tendencies associated with broad personality dimensions." Psychology and Psychotherapy: Theory, Research and Practice 77, no. 3: 397-402.

Haslam, N., T. Reichert, and A. P. Fiske 2002. Aberrant Social Relations in the Personality Disorders. Psychology and Psychotherapy: Theory, Research and Practice 75:19–31.

(See also: Brito, Rodrigo, Sven Waldzus, Maciej Sekerdej, and Thomas Schubert 2011. The contexts and structures of relating to others: How memberships in different types of groups shape the construction of interpersonal relationships. Journal of social and personal relationships 28, no. 3: 406-432.)

Reviewer #3: Thank you for the opportunity to review this manuscript!

I was very pleased to see that someone has devoted himself to the topic of measuring relational models, after the research here has to fall back (by necessity) on a few scales that do not always "work" optimally.

I find your work overall very promising and of practical value for future research on RMT. However, below are a number of comments that might help you make the article even more informative for the reader.

Introduction:

The explanation of Relational Models Theory seems to me to be a bit brief and I am not sure whether a reader who is not already familiar with this theory will really understand from this brief description what exactly the core ideas are, what exactly relational models are, what distinguishes the theory from other theoretical frameworks of social interaction and why it is significant and promises added value compared to other explanatory models. My guess is that many readers are not at all familiar with the theory and cannot assess to what extent the present research of an instrument revision is at all relevant.

I think the manuscript would benefit greatly from expanding on this part. In which parts of psychological research is the theory applied? What phenomena can it explain? What are examples of recent studies in the different psychological sub-disciplines? A few recent studies (Keck et al., 2018, Vodosek, 2012) have already been cited, but not really described.

Maybe you could use recent studies to give the reader an impression that the theory (which is, after all, somewhat older and does not really receive much attention in many psychological sub-disciplines) is definitely still relevant, which is why it is worthwhile to further develop the measurement of relational models. My impression is that - especially in organizational psychology - there is even a growing interest in this theoretical framework to explain social interactions, both on a theoretical (e.g., Bridoux & Stoelhorst, 2016; Mossholder et al., 2011) and on an empirical level (e.g., Arendt et al., 2021; Keck et al., 2018; Stofberg et al., 2019). It should also be mentioned that relational models theory has undergone several " enhancements " in the past decades, namely in the form of Relationship Regulation Theory (Rai & Fiske, 2011) and Relational Incentives Theory (Gallus et al., 2021). Please do not misunderstand me: It is not necessary to provide a comprehensive overview of all empirical research, but the reader should be given some insight into recent work on relational models in order to enable him/her to assess the practical value of a revised scale.

Measuring relational Models

• It might be worth mentioning that Vodosek's (2009) scale, that you mention and cite, measures relational models at team level and not at individual level.

Against this background, the fact that the items did not or only insufficiently form a 4-factor structure is to be evaluated somewhat differently than in the case of a scale that refers to individual dyadic relationships.

Already when formulating the Relational Models Theory, Fiske explicitly emphasized that no relationship takes place exclusively in one model and that it is actually always a matter of mixed forms from different models. It is certainly the case that in many (dyadic) relationships a tendency towards certain models is recognizable (which is supported by the previous results). However, with a methodological approach and a scale that refers to the relationships in an entire work team, as is the case in Vodosek's studies, one is confronted with the problem that in a team with its various members, relationships, team fault lines, etc., there is considerably more heterogeneity with regard to the relational models that are applied than in dyadic relationships. I think that Vodosek's scale can therefore only be compared to a limited extent with scales that consider dyadic relationships.

There is also another adaption of the MORQ that is not yet mentioned in your manuscript: Biber et al. (2008) used a German adaption of the MORQ, that is provided by Hupfeld-Heinemann (2005). Interestingly this scale has also 20 items with 5 for each relational model. To my knowledge, the details of the scale construction and validation are only described in German but maybe the Author has also material in English. It would be interesting to know if his adaption of the scale resulted in the same or similar items as yours.

Results:

With regard to the results section and the scale-analytical results, I would like to note that although I am familiar with these topics in principle, I do not consider myself an expert in this area. Although the descriptions of the authors are coherent and comprehensible for me, I recommend to attach more importance to the judgment of the other reviewer(s) than to mine in this part of the work.

References

Arendt, J. F. W., Kugler, K. G., & Brodbeck, F. C. (2021). Conflicting relational models as a predictor of (in)justice perceptions and (un)cooperative behavior at work. Journal of Theoretical Social Psychology, 5, 183-202. https://doi.org/10.1002/jts5.85

Biber, P., Hupfeld, J., & Meier, L. L. (2008). Personal values and relational models. European Journal of Personality, 22(7), 609-628.

Bridoux, F., & Stoelhorst, J. W. (2016). Stakeholder relationships and social welfare: A behavioral theory of contributions to joint value creation. Academy of Management Review, 41(2), 229-251. https://doi.org/10.5465/amr.2013.0475

Gallus, J., Reiff, J., Kamenica, E., & Fiske, A. P. (2021). Relational incentives theory. Psychological Review, Advance online publication. https://doi.org/10.1037/rev0000336

Hupfeld-Heinemann, J. (2005). Die grammatik sozialer beziehungen [Habilitation Thesis, Universität Bern]. Bern.

Keck, N., Giessner, S. R., Quaquebeke, N., & Kruijff, E. (2018). When do followers perceive their leaders as ethical? A relational models perspective of normatively appropriate conduct. Journal of Business Ethics. https://doi.org/10.1007/s10551-018-4055-3

Mossholder, K. W., Richardson, H. A., & Settoon, R. P. (2011). Human resource systems and helping in organizations: A relational perspective. The Academy of Management Review, 36(1), 33-52. https://doi.org/10.5465/amr.2009.0402

Rai, T. S., & Fiske, A. P. (2011). Moral psychology is relationship regulation: Moral motives for unity, hierarchy, equality, and proportionality. Psychological Review, 118(1), 57-75. https://doi.org/10.1037/a0021867

Stofberg, N., Bridoux, F., Ciulli, F., Pisani, N., Kolk, A., & Vock, M. (2019). A relational‐models view to explain peer‐to‐peer sharing. Journal of Management Studies. https://doi.org/10.1111/joms.12523

Vodosek, M. (2009). The relationship between relational models and individualism and collectivism: Evidence from culturally diverse work groups. International Journal of Psychology, 44(2), 120-128. https://doi.org/10.1080/00207590701545684

6. PLOS authors have the option to publish the peer review history of their article (what does this mean?). If published, this will include your full peer review and any attached files.

Reviewer #1: No

Reviewer #2: **Yes: **Alan P Fiske

Reviewer #3: No

---

## [Author Response · Author response to Decision Letter 0]

18 May 2023

Dear Editor

Thank you for your review and thoughtful comments from reviewers. We enclose a revised version of the manuscript that addresses the points raised during the review process.

Editor's Comment: Your manuscript requires minor revisions, and its acceptance is conditional on addressing all comments - which are rather minor- raised by all the reviewers. Please note that you do not have to reference all the papers suggested by reviewer 3.

Response: Thank you for this encouraging and positive evaluation. We have addressed all comments (itemised below) along with a marked-up copy of your manuscript highlighting changes made to the original version called 'Revised Manuscript with Track Changes'. Along with an unmarked version of your revised paper without tracked changes as a separate file labelled 'Manuscript'.

Reviewer 1 comments:

Comment: This manuscript presents five studies that attempt to clarify the structure of interpersonal relationships as assessed by a self-report measure based on relational models theory (RMT). The studies compellingly show that although there is good evidence of four relationship factors consistent with RMT, the existing measure of those factor is structurally weak. More importantly, they show that the measure can be substantially improved by (a) removing poor items, (b) adding a nonspecific factor capturing relationship closeness (or something similar), and (c) allowing one correlation between two factors (in place of the six included in earlier structural models).

Collectively the studies make several worthwhile contributions. They 1) produce a substantially improved measurement scale that will be useful for researchers interested in RMT; 2) affirm its four-dimensional structure; and 3) provide a helpful and simplifying theoretical account of why model fit of the original scale was poor, namely that it failed to consider generic closeness of relationships separate from their underlying model. In essence, we can understand and measure relationships in terms of the model or models on which they are based, AND on their level of closeness or engagement.

Methodologically the studies are strong. Samples are consistently larger than prior research, structural analyses are carried out carefully and with sophistication, and replication with independent samples is consistently sought and demonstrated.

Response: We thank you for the acute summary and positive comments!

In addition to these positive comments I have a few small concerns and comments.

Comment 1. On line 35, where it says "Relational models theory (RMT) offers a comprehensive classification of interpersonal relationships" it might be better to replace "classification" with "model" or something to avoid the mistaken impression that any relationship must belong to a single model.

Response: We replaced 'classification' with 'model'. (page 3, line 37)

Comment 2. On line 89, where it says "A two-factor model consisting of orthogonal bipolar dimensions, one running from AR to EM and one from CS to MP", it could be noted what these dimensions mean (e.g., equal-unequal and close-distant). The context here is that whereas RMT proposed four factors, two-dimensional structures like this are popular in social psychology.

Response: Thank you, we agree that this section was unclear: We now say "one running from EM to AR and one from CS to MP, thus capturing the equality-inequality and closeness-distance dimensions" (page 7, lines 149-150).

Comment 3. The Introduction focuses on the issue of how to measure the RMs, without giving even a brief account of the evidence that they matter, such as referring to empirical work showing they predict things that matter. I don't think a full literature review is needed here, but perhaps some citation or work showing the value of the RMT approach would help justify the effort to improve their measurement.

Response: Thank you. Reviewer 3 made a similar comment. We have expanded the introduction by a page, expanding more on the value and strengths and application of RMT, including a section on the predictive validity of RMT: 

" A substantial amount of empirical research has demonstrated that relational models can accurately predict significant outcomes. For instance, Vodosek [9] found that horizontal collectivism was associated with equality matching and communal sharing relationships, whereas vertical individualism was related to a preference for authority ranking, and vertical collectivism was related to a preference for authority ranking and communal sharing. Biber et al. [10] investigated the relationship between relational models and universal human values [11]. They found that individuals who prioritise CS relationships place greater importance on benevolence and universalism values while placing less emphasis on power and achievement. Conversely, those who value AR or MP relationships tend to prioritise power and achievement values over benevolence and universalism. A disparity between anticipated and real relationship models resulted in a sense of inequity among employees at work [12, 13], and they began to view their supervisors as lacking morals [8]. In clinical samples, different diagnoses were linked to either difficulties or extreme use of specific relational models [14, 15]. For instance, dysthymia was found to be positively associated with high levels of AR relationships with close friends and family members, while hypomania was positively associated with high levels of CS and EM relationships with authority figures." (See pages 4-5, lines 80-95).

Comment 4. A related issue is that the manuscript is present[ed] mainly as being about improving a self-report measure and determining its structure, but isn't it also saying something about the actual structure of relationships? Maybe the authors could add a brief statement about whether they think their findings tell us something about how relationships are structured, or do they only tell us about one questionnaire?

Response: Thank you for the comment. While our study aimed to improve a self-report measure of social relationships and our findings were limited to the questionnaire we used, we do believe that our findings shed light on the underlying structure of social relationships. We now say (pages 25-26, lines 567-601).

" Although our manuscript primarily aimed to enhance the MORQ's psychometric properties and confirm its four-factor structure, our findings also provide insights into the underlying structure of social relationships. Our results suggest that relationships are structured around, at a minimum, these four models of interpersonal relations. Moreover, our findings refute the notion that these four models are merely consequences of a simpler, two-dimensional model (such as equality-inequality or close-distant) since these models did not adequately fit our data." (See page 27, lines 587-592)

Comment 5. On line 218 it says "it may be necessary to replace correlations between factors with a general relationship factor". A brief comment about what such a "general factor" is and what finding evidence of it would mean might help here.

Response: Thank you, we now clarify this by adding "…factor, representing a general tendency to initiate or avoid relationships with other people". (See page 13, lines 297-298). We also pick this up in the general discussion.

Comment 6. Line 267 says "which did not replicate well in the hold-out relationship datasets". This comes across as a slight exaggeration: in the replications the CFI is only worse by 3.4% on average and the TLI by 4.1%. Yes there is a drop off, but this is something less than a complete failure to replicate.

Response: We agree that the size of a drop-off was not sufficient to call it a failure. We now say instead: "While the drop-off was not substantial, we wished to investigate whether more complex models would reliably yield a good fit" (page 17, lines 356 - 357)

Comment 7. On line 284, where it says "Fig 2 shows the final model (using the dataset with all five relationships combined)", it's unclear what the combined dataset means. Are the five relationships for each individual treated as independent or is between-subjects variance controlled for? I felt this dataset wasn't explained adequately.

Response: We made the dataset description clearer by saying: "Fig 2 shows the final model (using the dataset with all five relationships combined after controlling for between-subject variance)" (page 17, lines 367-368)

Comment 8. On line 297, where it says "We also found that the model required a general factor loading on all items, which was not found in previous studies", it should also be noted that it wasn't found because it wasn't looked for. Past confirmatory studies couldn't have found this factor because they were specifying four RMT factors only.

Response: Thank you for this correction. We now say: "We also found that the model required a general factor loading on all items, which was not investigated in previous studies relying on factor correlations only." (page 18, lines 379-380)

Comment 9. On line 344, "AM" should be "AR".

Response: Thank you for catching this typo. It is now corrected (page 20 line 428)

Comment 10. On line 407 it looks very odd to have three correlations in a row that are precisely 0.00.

Response: Thank you: We had rounded these values to too few decimal places. However, we were also including all five responses rather than just one in our correlation. Fixing this does not alter the outcomes materially, and the relationship remains not significant (see page 23, lines 502-508).

Comment 11. On line 421 and figure 4 it is very interesting that "employee" is much higher on the factor than "employer/manager". This suggests the factor is not about CS/EM/MP versus the AR model in general, as the factor loadings might seem to suggest, but that it's the person's position within the authority relationship that matters. Unless I am misunderstanding it, it seems employers report feel more close to or engaged with their employees than vice versa. This point might merit a brief exploration in the Discussion, as it helps to make the point that the general factor - though the study doesn't entirely nail down what it is - is not just about authority or inequality versus its absence.

Response. We agree with the reviewer that the difference between the general factor scores on employer and employee relationship requires further clarification. We added the following to a discussion: "The employee relationship had a much higher score on the general factor than the employer relationship, despite both representing the AR relationship. We believe this may reflect the paternalistic side of authoritarian leadership in which an employer's role involves, to a degree at least, responsibility towards their employees" (see page 25, lines 535-538 )

Reviewer 2 comments

This paper reports a research program aiming to replicate and improve the MORQ, a measure of the relational models proposed by RMT. The paper is cogent and clearly written. The series of studies is well conceived, and the results are valuable.

The first two studies (1a, 1b) report basically a failed smaller scale effort, before Studies 2 and 3 provide a better and stable solution with more data. Study 4 explores correlation of observed variables with additional measures. One might wonder whether #1 should then be still reported, but I agree that it is good to report it to see the full development. 

Response: Thank you for this positive evaluation and accurate summary!

Comment 1. The only thing I would suggest in addition is to test the final model again on the data from Study 1 to provide another internal replication.

Response: Sadly, the data we collected in study 1 have only one relationship reported per respondent, meaning we cannot control for respondent variance in these data and, therefore, cannot use that dataset for internal replication. But we were greatly boosted in our confidence in the model when the model replicated without change in study 3 (an independent sample).

Comment 2 It is very commendable that all data from this study are available. Both this paper and the data will be an excellent resource for future work on the MORQ.

Response: Thank you: We hope others download and use these open data.

Comment 3. One of the many merits of these studies is that the participants were UK residents, in contrast to the participants in the original studies of Haslam and Fiske. Thus the current findings constitute a cross-cultural replication, albeit in a very closely related culture. This should be mentioned, along with the fact that three decades have elapsed since the original studies. These two differences may partially account for the finding that some of the original items did not load on the expected factors.

Response: We now acknowledge this by saying: " The present studies also were conducted thirty years after the initial study and in a different yet related culture (the UK compared to the US). This may partially account for the finding that some original items did not load on the expected factors. The finding that despite the three decades having elapsed and testing in a much changed and different culture in the UK, the model was validated is a testimony to the durability of the RMT model." (See page 28, lines 602-607)

Comment 4. On p. 9 the authors write that "following Haslam and Fiske (3) between-participant variance was controlled by regression participant ID on all item responses". Haslam & Fiske (1999) did this: "In each analysis, one item was regressed on 41 dummy variables collectively representing the 42 participants, and the unstandardised residuals were retained.". Did the current paper do the same? The sentence above is vague.

Response: We now made it clearer and say that " Before conducting inferential analyses, following Haslam and Fiske (3), the impact of reporter-specific variance in the multiple target reports from each subject was controlled. Where Haslam and Fiske accomplished this by dummy coding the participant IDs and residualising the data for these dummy variables, we accomplished the same purpose in a multi-level analysis, with participant ID as a random variable, again retaining the unstandardised residuals." (See page 13, lines 276-280 )

Comment 5. The method applied by Haslam & Fiske (1999) seems somewhat outdated. Wouldn't it be better to run the confirmatory factor analysis on a multilevel model, where random variance due to participants is added in the form of a random factor? Is that available in the R packages used here?

Response: We agree (see our response to Comment 4) – we apologise for not making this clear in the original methodology.

Comment 6. The holdout samples in Study 2 fail to replicate the structure according to the criteria set earlier – CFI and TLI > .95, RMSEA < .06. But I noted in Table 2 that RMSEA actually holds up well, it's the other two indices that are poor. Can this be interpreted? More generally, it would be helpful to the general reader if the authors wrote a few sentences explain what each of the indices assesses, and what the conventions are for acceptable scores, and good scores.

Response: If you referring to Table 3, we agree, but then we added a general factor which improved the model fit, shown in Table 4 (We have also updated the table captions for clarity). If you referring to Table 4 (for the final model), we think that the final fit indices indicate replication, rather than a failure to replicate in the hold out samples. The RMSEA as you say was good and improved in most holdouts, and the TLI/CFI remained adequate or improved in nearly all cases. We do note this now saying: "… which also replicated well across the holdout datasets. The RMSEA remained below the threshold in all tests while other fit indices improved or decreased in proportion as compared to the fit obtained in the initial data " (page 17, lines 364-367)

And regarding the fit indices, we added: " Model fit was assessed using the Comparative Fit Index (CFI), Tucker-Lewis Index (TLI), and the root mean square error of approximation (RMSEA). The RMSEA evaluates the deviation of a hypothesised model from an ideal one. It ranges between 0 and 1, with values closer to zero indicating a better fit. In contrast, the CFI and TLI compare the fit of a hypothesised model to that of a baseline model, which assumes no correlation between any underlying continuous variables. Higher values, closer to 1.0, indicate a better fit for CFI and TLI. Following Hu & Bentler [23] and Yu [24], we adopted criteria of TLI and CFI >= .95 and RMSEA <= .06” (See Pages 9-10, lines 212- 218)

Comment 7. On p. 19, the Discussion includes the sentence "Although we controlled between-participant variance by regressing participant ID on all item responses, these replication datasets were thus not independent." – This was surprising, is it a copy-paste mistake? Why would you do that procedure if the data are not analysed in the same model? If you analyse the datasets separately, this regression shouldn't do anything, and seems unnecessary. Please clarify.

Response: This was perhaps a confusing presentation, unnecessarily mixing the multilevel control with the need for replication, which was our only concern. We now say: " While the model replicated in the different target data provided by our subjects, we wished to replicate the model in a completely independent dataset to further corroborate the new model. We, therefore, conducted Study 3, testing the exact final model from Study 2 in a new dataset." See page 18, lines 383-386

Comment 8. On p. 16, you say "… that this is the true model of MORQ" – I would object to this way of putting it. You found a model that fits the data well, as well as new data. No model is ever true; a model is simply a model! The only question is whether it becomes useful by approximating the data.

Response: We agree and now say "reliable, well-fitting, and useful model of MORQ" (page 20, line 425)

Comment 9. Study 4 matched new data to the dataset from Studies 2 and 3. This seems to contradict the earlier statements that "No personal identifying information was collected". I assume that you matched participants using their Prolific ID? If so, you should clarify that the researcher team could not identify the participants. Please make it clear how you were able to match. 

Response: We collected prolific ID's (this is done automatically by the Prolific platform) and used them to merge the datasets. We now say " All data were de-identified and collected using Prolific IDs to protect participants' privacy. No personally identifying information was collected and the authors did not have access to information that could identify individual participants during or after data collection. For privacy, Prolific IDs have been anonymised and replaced with numerical IDs in the open data associated with this manuscript." (Page 9, lines 207-211)

Comment 10. Related: I actually had a look at the Study 2 dataset available on OSF and did not see any ID variable for participants, just wave, gender, and age. That should be added?

Response: We removed prolific participants' IDs from the publicly shared dataset as this information can be seen as sensitive. As per your request, we have now anonymised the Prolific IDs and replaced them with numerical IDs to ensure reproducibility while preserving the confidentiality.

Comment 11. p. 19: "the general factor and RWA were unrelated (r(2193)" – I couldn't figure out how this was calculated. If this is a simple correlation, how did you arrive at a variable representing the general factor? Or was this done in the model, and we just see the estimate from the model? But then how do we get an r? Again, wouldn't this be better done in a multilevel model? Same question for the remaining analyses in Study 4.

Response: We used factor scores extracted from the model. We clarified this in text: " First, we tested the hypothesis that authoritarianism explains the general factor scores extracted from the model using the umx function umxFactorScores(). Regression scoring was used to determine the factor scores, and potential confounding effects of authoritarianism were tested in each of the five relationships examined." (See page 23, lines 496-499)

Comment 12. The final model features a general "general relationship" factor, which however doesn't load (or even loads negatively) on the AR items. In addition, the model fit requires a rather high correlation of the latent CS and EM variables. The discussion already provides a good start at explaining these. It's not fully satisfying, but that's ok. My question is however: What are the recommendations for future work with the MORQ (in particular the item selection here)? Should researchers simply average items to scales and proceed as usual? Or would it be better to replicate the observed structure, and then either work with the latent variables, or somehow save the (predicted?) values?

Response: For future work with MORQ, we would suggest developing an extended questionnaire to replace the missing items. For the short version of MORQ developed here, we recommend scoring by averaging (or summing) responses. We clarified that now by saying: " As the original 33 items were designed to cover a spectrum of behavioural domains, generating new items to replace the missing items may be of value to capture the complete spectrum of relationship models. That said, the scales developed here and scored by averaging responses for each scale should be valid for their intended purposes or for identifying relationship models." (See page 27, lines 595-599).

Comment 13. I was a bit surprised and perplexed by how speedily participants completed the tasks in each of the studies. Do we know that all participants conscientiously completed all tasks? Was there any check on that?

Response: We tested the questionnaires before releasing them to the participants, and the mean completion times were within a reasonable margin. Participants received the completion code at the end of the questionnaire, and all items were coded as 'forced response' (e.g. participants were not able to skip items). 

Comment 14. On lines 44-45, the authors write that Fiske posited the four relational systems to be the dimensions of social relationships. That is definitely not what Fiske posited; he posited that the relational models are four distinct structures of relationships, and not dimensions. Haslam's subsequent work, using several taxometric methods, supported that conclusion. Regarding the dimensions vs. categories question: this is a bit hard to ask, because H&F1999 doesn't really discuss this ether, despite the fact that H1994, using data collected with the same instrument, already shows that people perceive their own relationship as categorical. So I would just invite the authors to ponder this.

Response: Thank you for this correction, we now use " four distinct relational systems constituting the structures of social relationships " instead of "four dimensions". (See page 3, lines 46 - 47)

Comment 15. Lines 45-46 (see also line 69), the authors write that authority ranking entails "respecting and obeying." In his 1991 book, his 1992 article, and subsequent publications he specifically wrote that command and obedience are not necessary features of authority ranking. In some authority ranking systems, command and obedience are among the cultural implementations of authority ranking, in other cultural implementations, not. If US News and World Report ranks university A above university B, that does not entail anything about commands or obedience. Buddhists regard the Buddha as the highest ranking being, while for Tibetan Buddhists, the Dali Lama is not far below the Buddha. But neither is entitled by their supreme rank to give orders to anyone.

Response: Thank you for this correction, we removed the 'obeying' statement. (See page 4, line 78)

Comment 16. Line 260 has a typo: "sets of items would permit fit this structural model." Line 344, "AM" should be 'AR.'

Response: Thank you for catching this typo. It is now corrected (page 20 line 428)

Comment 17. I wonder whether the "general factor" assesses commitment and devotion to the relational model, where the low end of the scale approaches what Fiske defines as 'Null relationships.'

Response: Thank you for this suggestion, we now say: "…this factor may assess general commitment and devotion to form social relationships with others, with low scores representing a “null relationship” [2], an avoidant attitude towards forming relationships." (page 20, lines 429-431)

Comment 18. Lines 495-6 the authors mention the value of a measure of the person's proclivity to form relationships based on the four respective relational models. Such a measure exists, and results from studies using it have been published. In essence, after a participant reports 40 relationships, the participant codes how well each one fits the four relational models, and the researcher simply counts how many of the person's relationships are reportedly structured according to each of the four respective rms. Here are a couple of references:

Allen, Nicholas B., Nick Haslam, and Assaf Semedar 2005. Relationship patterns associated with dimensions of vulnerability to psychopathology. Cognitive Therapy and Research 29: 733-746.

Caralis, Dionyssios, and Nick Haslam 2004. "Relational tendencies associated with broad personality dimensions." Psychology and Psychotherapy: Theory, Research and Practice 77, no. 3: 397-402.

Haslam, N., T. Reichert, and A. P. Fiske 2002. Aberrant Social Relations in the Personality Disorders. Psychology and Psychotherapy: Theory, Research and Practice 75:19–31.

(See also: Brito, Rodrigo, Sven Waldzus, Maciej Sekerdej, and Thomas Schubert 2011. The contexts and structures of relating to others: How memberships in different types of groups shape the construction of interpersonal relationships. Journal of social and personal relationships 28, no. 3: 406-432.)

Response: Thank you for these useful suggestion, we now include the relationship profile scale as one of the developments of the RMT: “First, a personality assessment tool – the Relationship Profile Scale [15], was developed to evaluate individual preferences for distinct relational models, measuring the perceived importance, satisfaction, challenges, and motivations associated with each of the four relational models. Together with the MORQ, the Relationship Profile Scale enables a comparison of individuals' desired and actual relationship experiences.” ( see page 5 lines 105-109)

Reviewer 3 comments:

 Thank you for the opportunity to review this manuscript!

I was very pleased to see that someone has devoted himself to the topic of measuring relational models, after the research here has to fall back (by necessity) on a few scales that do not always "work" optimally.

I find your work overall very promising and of practical value for future research on RMT. However, below are a number of comments that might help you make the article even more informative for the reader.

Response: Thank you for the very positive feedback!

Introduction:

Comment 1. The explanation of Relational Models Theory seems to me to be a bit brief and I am not sure whether a reader who is not already familiar with this theory will really understand from this brief description what exactly the core ideas are, what exactly relational models are, what distinguishes the theory from other theoretical frameworks of social interaction and why it is significant and promises added value compared to other explanatory models. My guess is that many readers are not at all familiar with the theory and cannot assess to what extent the present research of an instrument revision is at all relevant.

Response: This broad comment was also reflected in Reviewer 1's comments. We have added around a page of background for the reader who is not already familiar with this theory so that they can grasp the relevance the present research. We agree that this direction benefits the paper greatly and hope we have expanded on this part sufficiently. 

Regarding the core ideas we say: “Based on ethnographic fieldwork and a review of previous studies, Fiske [1] proposed four distinct relational systems constituting the structures of social relationships. These four models are theorised as fundamental and innate and serve as a comprehensive framework to describe all possible human relationships [6]. They depict how individuals evaluate their status in relation to others and elucidate appropriate or inappropriate behaviours in a given social context. In essence, they offer a framework for comprehending social interactions and the expected norms of behaviour in diverse social settings.“(Page 3, lines 46-52)

Regarding what the models are, we improved our description of the relational models and say:

“The first of these models, Communal Sharing (CS), focuses on what people have in common and is exemplified in relationships where people share an identity with others, such as family, tribe, religion, or ethnic group, resulting in mutual recognition of social equivalence of individuals. This shared identity is reflected in helping others regardless of their past contributions, treating the property as communal, and making joint consensus-based decisions. The second relational model is Equality Matching (EM), in which individuals treat each other as distinct but equal partners. In EM relationships, work inputs and outputs are divided equally where possible. Where resources and work are not divisible equally, individuals keep a count of what they give and receive and equalise this over time. Examples of this relationship include mutual credit organisations and babysitting co-ops. EM also extends to vengeful behaviour, such as eye-for-an-eye justice [7]. The third model, Authority Ranking (AR), implements a hierarchy system in which social interactions are based on recognising and respecting different levels of authority. The distribution of resources in this model is expected to be unequal, with superiors feeling entitled to a larger share of resources and subordinates accepting this division as fair [8]. A range of factors can influence ranking in an AR, including age, gender, seniority, and achievement. One example of this model would be the relationship between employer and employee. The fourth and final relational model in RMT is Market Pricing (MP). The MP model suggests that people relate to each other based on the value they exchange in a relationship as if it were a market transaction. According to this model, individuals perceive their relationships as a means to obtain desired resources, assistance, or support from the other person. Examples of relationships that align with this model include commercial partnerships, where transactions are prominent, as well as cultural constructs like the concepts of price, wages, or dividends.” (Pages 3-4, lines 52-74)

Regarding what distinguishes RMT from other social interactionist accounts we say:

“Perhaps the key feature distinguishing RMT from theories of social relationships, such as interdependence theory [16], attachment theory [17] and social identity theory [18] is the emphasis RMT places on explaining the underlying structure of relationships. Rather than focusing on the role of interdependence within relationships, emotional bonds formed early in life or the sense of self derived from a social group membership, RMT provides a framework for understanding social interactions and the appropriate behaviours within them.” (Page 5, lines 98-103)

Finally, regarding empirical studies showing the validity of RMT we now say: " A substantial amount of empirical research has demonstrated that relational models can accurately predict significant outcomes. For instance, Vodosek [9] found that horizontal collectivism was associated with equality matching and communal sharing relationships, whereas vertical individualism was related to a preference for authority ranking, and vertical collectivism was related to a preference for authority ranking and communal sharing. Biber et al. [10] investigated the relationship between relational models and universal human values [11]. They found that individuals who prioritise CS relationships place greater importance on benevolence and universalism values while placing less emphasis on power and achievement. Conversely, those who value AR or MP relationships tend to prioritise power and achievement values over benevolence and universalism. A disparity between anticipated and real relationship models resulted in a sense of inequity among employees at work [12, 13], and they began to view their supervisors as lacking morals [8]. In clinical samples, different diagnoses were linked to either difficulties or extreme use of specific relational models [14, 15]. For instance, dysthymia was found to be positively associated with high levels of AR relationships with close friends and family members, while hypomania was positively associated with high levels of CS and EM relationships with authority figures." (See pages 4-5, lines 80-95).

Comment 2: It should also be mentioned that relational models theory has undergone several " enhancements " in the past decades, namely in the form of Relationship Regulation Theory (Rai & Fiske, 2011) and Relational Incentives Theory (Gallus et al., 2021). Please do not misunderstand me: It is not necessary to provide a comprehensive overview of all empirical research, but the reader should be given some insight into recent work on relational models in order to enable him/her to assess the practical value of a revised scale.

Response: Thank you for this suggestion, we now mention this in the introduction, saying:

" RMT has continued to evolve and expand its realm of application, with several changes being of particular relevance. First, a personality assessment tool – the Relationship Profile Scale [15], was developed to evaluate individual preferences for distinct relational models, measuring the perceived importance, satisfaction, challenges, and motivations associated with each of the four relational models. Together with the MORQ, the Relationship Profile Scale enables a comparison of individuals' desired and actual relationship experiences. A significant theoretical advance known as Relationship Regulation Theory [19] extended RMT into the domain of moral psychology by associating each relational model with four distinct moral motives. For instance, the moral motive of hierarchy is based on the AR relationship and its focus on establishing and upholding a clear ranking in social groups. The motive of hierarchy motivates those in lower positions to show respect, obedience, and deference to those above them, including leaders, ancestors, or gods, and to punish those who go against them. Conversely, those in higher positions feel a moral responsibility to guide, direct, and safeguard those below them. This expansion links RMT to existing moral theories [20, 21]) but construes the nature of moral behaviour as relationship management and emphasises that the moral value of acts such as harming, unequal treatment, or being impure are dependent on the relationships and relational models within which they are deployed. 

Most recently, RMT has undergone another significant enhancement by incorporating the well-established effects of incentives on behaviour into our understanding of relationships and relationship management. Known as Relational Incentives Theory [22], this extension posits that for incentives to be effective, they should align with relational models. For instance, incentives promoting communal sharing relations should be most effective when they align with the motive of unity, while proportional incentive schemes work best for market pricing relations. These recent advancements demonstrate the continued significance of RMT and highlight the crucial role of the four relational models in comprehending and predicting diverse behaviours, ranging from resolving moral disagreements to determining the efficacy of incentive schemes. 

.” See Pages 5-6, lines 104-129 

Comment 3: It might be worth mentioning that Vodosek's (2009) scale, that you mention and cite, measures relational models at team level and not at individual level.

Against this background, the fact that the items did not or only insufficiently form a 4-factor structure is to be evaluated somewhat differently than in the case of a scale that refers to individual dyadic relationships.

Already when formulating the Relational Models Theory, Fiske explicitly emphasised that no relationship takes place exclusively in one model and that it is actually always a matter of mixed forms from different models. It is certainly the case that in many (dyadic) relationships a tendency towards certain models is recognisable (which is supported by the previous results). However, with a methodological approach and a scale that refers to the relationships in an entire work team, as is the case in Vodosek's studies, one is confronted with the problem that in a team with its various members, relationships, team fault lines, etc., there is considerably more heterogeneity with regard to the relational models that are applied than in dyadic relationships. I think that Vodosek's scale can therefore only be compared to a limited extent with scales that consider dyadic relationships.

Response: We agree with this possibility and now mention that Vodosek's (2009) scale measures relational models at team level (page 6, lines 119-121):

 "In this study, however, participants were asked to indicate the degree to which they believed each MORQ statement should be true in an ideal working group, rather than rating their actual relationships. This difference in approach may account for the low level of fit observed." (See Page 8, lines 164-165)

Comment 4: There is also another adaption of the MORQ that is not yet mentioned in your manuscript: Biber et al. (2008) used a German adaption of the MORQ, that is provided by Hupfeld-Heinemann (2005). Interestingly this scale has also 20 items with 5 for each relational model. To my knowledge, the details of the scale construction and validation are only described in German but maybe the Author has also material in English. It would be interesting to know if his adaption of the scale resulted in the same or similar items as yours.

Response: Thank you for this interesting suggestion. We have now emailed the corresponding author, Jorg Hupfeld, requesting the English items they have chosen for their scale.

Comment 5: With regard to the results section and the scale-analytical results, I would like to note that although I am familiar with these topics in principle, I do not consider myself an expert in this area. Although the descriptions of the authors are coherent and comprehensible for me, I recommend to attach more importance to the judgment of the other reviewer(s) than to mine in this part of the work.

Response: Thank you for your positive evaluation.

References

Arendt, J. F. W., Kugler, K. G., & Brodbeck, F. C. (2021). Conflicting relational models as a predictor of (in)justice perceptions and (un)cooperative behavior at work. Journal of Theoretical Social Psychology, 5, 183-202. https://doi.org/10.1002/jts5.85

Biber, P., Hupfeld, J., & Meier, L. L. (2008). Personal values and relational models. European Journal of Personality, 22(7), 609-628.

Bridoux, F., & Stoelhorst, J. W. (2016). Stakeholder relationships and social welfare: A behavioral theory of contributions to joint value creation. Academy of Management Review, 41(2), 229-251. https://doi.org/10.5465/amr.2013.0475

Gallus, J., Reiff, J., Kamenica, E., & Fiske, A. P. (2021). Relational incentives theory. Psychological Review, Advance online publication. https://doi.org/10.1037/rev0000336

Hupfeld-Heinemann, J. (2005). Die grammatik sozialer beziehungen [Habilitation Thesis, Universität Bern]. Bern.

Keck, N., Giessner, S. R., Quaquebeke, N., & Kruijff, E. (2018). When do followers perceive their leaders as ethical? A relational models perspective of normatively appropriate conduct. Journal of Business Ethics. https://doi.org/10.1007/s10551-018-4055-3

Mossholder, K. W., Richardson, H. A., & Settoon, R. P. (2011). Human resource systems and helping in organisations: A relational perspective. The Academy of Management Review, 36(1), 33-52. https://doi.org/10.5465/amr.2009.0402

Rai, T. S., & Fiske, A. P. (2011). Moral psychology is relationship regulation: Moral motives for unity, hierarchy, equality, and proportionality. Psychological Review, 118(1), 57-75. https://doi.org/10.1037/a0021867

Stofberg, N., Bridoux, F., Ciulli, F., Pisani, N., Kolk, A., & Vock, M. (2019). A relational‐models view to explain peer‐to‐peer sharing. Journal of Management Studies. https://doi.org/10.1111/joms.12523

Vodosek, M. (2009). The relationship between relational models and individualism and collectivism: Evidence from culturally diverse work groups. International Journal of Psychology, 44(2), 120-128. https://doi.org/10.1080/00207590701545684

Comment 1. Please ensure that your manuscript meets PLOS ONE's style requirements, including those for file naming. The PLOS ONE style templates can be found at

Comment 2. Please change "female" or "male" to "woman" or "man" as appropriate, when used as a noun (see for instance https://apastyle.apa.org/style-grammar-guidelines/bias-free-language/gender).

Response: We corrected the noun usage (see Page 6, line 132; page 9, line 193; page 11, line 246; page 15, line 327)

Comment 3. Please provide additional details regarding ethical approval in the body of your manuscript. In the Methods section, please ensure that you have specified the name of the IRB/ethics committee that approved your study.

Response: We have specified the ethics committee name in methods (see page 6, lines 134-135; page 9, lines 197-198; page 12, lines 251-252; page 16, line 334; page 19, line 399)

Comment 4. Please provide additional details regarding participant consent. In the ethics statement in the Methods and online submission information, please ensure that you have specified what type you obtained (for instance, written or verbal, and if verbal, how it was documented and witnessed). If your study included minors, state whether you obtained consent from parents or guardians. If the need for consent was waived by the ethics committee, please include this information.

Response: No minors participated in the study. We specified the type of consent obtained in the ethics statement in each study (see page 9, lines 199-200; page 12, line 264; page 15, line 322; page 22, line 472)

Comment 5. Please include your full ethics statement in the 'Methods' section of your manuscript file. In your statement, please include the full name of the IRB or ethics committee who approved or waived your study, as well as whether or not you obtained informed written or verbal consent. If consent was waived for your study, please include this information in your statement as well.

Response: We provided this information; see our response to comments 4 and 5.

Comment 6. Please include captions for your Supporting Information files at the end of your manuscript, and update any in-text citations to match accordingly. Please see our Supporting Information guidelines for more information: http://journals.plos.org/plosone/s/supporting-information.

Response: We now added captions.

Comment 7. Please review your reference list to ensure that it is complete and correct. If you have cited papers that have been retracted, please include the rationale for doing so in the manuscript text, or remove these references and replace them with relevant current references. Any changes to the reference list should be mentioned in the rebuttal letter that accompanies your revised manuscript. If you need to cite a retracted article, indicate the article's retracted status in the References list and also include a citation and full reference for the retraction notice.

Response: We have checked the references list. We have not cited any papers that have been retracted.

---

## [Decision Letter · Decision Letter 1]

5 Jun 2023

Relational Models Theory: Validation and replication for four fundamental relationships.

PONE-D-22-34162R1

Dear Dr. Zakharin,

We’re pleased to inform you that your manuscript has been judged scientifically suitable for publication and will be formally accepted for publication once it meets all outstanding technical requirements.

Kind regards,

Srebrenka Letina, Ph.D.

Academic Editor

PLOS ONE

Additional Editor Comments (optional):

Reviewers' comments:

Reviewer's Responses to Questions

**Comments to the Author**

1. If the authors have adequately addressed your comments raised in a previous round of review and you feel that this manuscript is now acceptable for publication, you may indicate that here to bypass the “Comments to the Author” section, enter your conflict of interest statement in the “Confidential to Editor” section, and submit your "Accept" recommendation.

Reviewer #1: (No Response)

Reviewer #2: All comments have been addressed

2. Is the manuscript technically sound, and do the data support the conclusions?

Reviewer #1: Yes

Reviewer #2: Yes

3. Has the statistical analysis been performed appropriately and rigorously? 

Reviewer #1: Yes

Reviewer #2: Yes

4. Have the authors made all data underlying the findings in their manuscript fully available?

Reviewer #1: Yes

Reviewer #2: (No Response)

5. Is the manuscript presented in an intelligible fashion and written in standard English?

Reviewer #1: Yes

Reviewer #2: Yes

6. Review Comments to the Author

Reviewer #1: The authors have done a careful job responding to my comments and I am completely happy with all of them with the partial exception of comment 5. I realize the meaning of the general factor hasn't been fully clarified by the studies and it would not be reasonable to expect it to be, but is it really about initiating versus avoiding relationships (lines 297-8)? This would seem to implying that the general factor is about relationship onset and quantity (i.e., whether and when they begin and how many the person has), not qualitative aspects of the relationships the person actually has (like closeness or intensity or level of engagement). Surely a tendency to avoid relationships would mean you have fewer of them rather than saying much about the the nature of the ones you have. The fact that the general factor is related to features of actual relationships (e.g., the degree to which they're CS, EM and MP but not AR) implies that the general factor is about HOW people conduct the relationships they have rather than a general tendency to start versus avoid relationships. Clearly that "how" isn't adequately captured by closeness, but what it is remains uncertain.

This is not a key issue and I don't think it has to be addressed, but I thought I'd raise it as something for the authors to consider and potentially tweak at lines 297-8 and potentially 576-585. To me, it would be better to be clear that any idea of the general relationship factor being about "avoidance" or "general commitment and devotion to form social relationships with others" has to do with how existing relationships are carried out (i.e., aspects of current relationships), not with whether they exist at all.

Reviewer #2: (No Response)

7. PLOS authors have the option to publish the peer review history of their article (what does this mean?). If published, this will include your full peer review and any attached files.

Reviewer #1: **Yes: **Nick Haslam

Reviewer #2: No

---

## [Editor Report · Acceptance letter]

8 Jun 2023

PONE-D-22-34162R1 

Relational Models Theory: Validation and replication for four fundamental relationships. 

Dear Dr. Zakharin:

I'm pleased to inform you that your manuscript has been deemed suitable for publication in PLOS ONE. Congratulations! Your manuscript is now with our production department. 

Kind regards, 

on behalf of

Dr. Srebrenka Letina 

Academic Editor

PLOS ONE